# Coating Thickness Determination Using X-ray Fluorescence Spectroscopy: Monte Carlo Simulations as an Alternative to the Use of Standards

**Walter Giurlani [1],\*** , **Enrico Berretti [2]**, **Massimo Innocenti [1,2]** and **Alessandro Lavacchi [2],\***

[1] Dipartimento di Chimica, Università degli Studi di Firenze, via della Lastruccia 3,
50019 Sesto Fiorentino (FI), Italy; m.innocenti@unifi.it

[2] Consiglio Nazionale delle Ricerche—Istituto di Chimica dei Composti OrganoMetallici (CNR-ICCOM),
via Madonna del Piano 10, 50019 Sesto Fiorentino (FI), Italy; eberretti@iccom.cnr.it

\* Correspondence: walter.giurlani@unifi.it (W.G.); alessandro.lavacchi@iccom.cnr.it (A.L.);
Tel.: +39-55-457-3102 (W.G.); +39-55-522-5250 (A.L.)

**Abstract:** X-ray fluorescence is often employed in the measurement of the thickness of coatings. Despite its widespread nature, the task is not straightforward because of the complex physics involved, which results in high dependence on matrix effects. Thickness quantification is accomplished using the Fundamental Parameters approach, adjusted with empirical measurements of standards with known composition and thickness. This approach has two major drawbacks: (i) there are no standards for any possible coating and coating architecture and (ii) even relying on standards, the quantification of unknown samples requires the precise knowledge of the matrix nature (e.g., in the case of multilayer coatings the thickness and composition of each underlayer). In this work, we describe a semiquantitative approach to coating thickness measurement based on the construction of calibration curves through simulated XRF spectra built with Monte Carlo simulations. Simulations have been performed with the freeware software XMI-MSIM. We have assessed the accuracy of the methods by comparing the results with those obtained by (i) XRF thickness determination with standards and (ii) FIB-SEM cross-sectioning. Then we evaluated which parameters are critical in this kind of indirect thickness measurement.

**Keywords:** X-ray fluorescence (XRF); thickness determination; thin film; simulation; XMI-MSIM; electrodeposition; Monte Carlo; galvanic industry; electroplating

## 1. Introduction

Thickness is a crucial parameter in coatings technology and affects material functionality. Thickness determination of metallic and ceramic coatings is often performed by X-ray Fluorescence (XRF), a widespread, non-destructive technique applied in the industry as a tool for Quality Assurance (QA) and Materials Science R&D [1–5]. Deriving a coating's thickness from the X-ray spectrum requires an experimental calibration curve that employs standards; however, due to the large dependence of the X-ray spectrum on the nature of the coating and the substrate, standards are not always available. The variability of thickness, layer composition, multilayer architectures, and substrate chemical nature create difficulties in producing certified standards. This issue is critical in industrial applications; among them, the determination of precious metal coatings in the fashion industry is a major one, as production employs a large number of coatings and substrates, with extreme variability in the system.

Nowadays, the most common approach is the use of the fundamental parameter (FP) method [6–9]. FP relies on a theoretical equation that considers the composition and thickness of the sample to evaluate the XRF intensity. Practically, the FP method is combined with a few empirical standards

to correct unpredicted deviations due to matrix effects [10,11]. With the FP method, it is possible to determine the film thickness of single and even multilayer samples if the structure and the composition are known exactly; nevertheless, the error correlated to the measurement is significant. Typical accuracy for single layer samples is ±5%, while for multiple-layer samples this value grows to ±10% for the upper layer and ±37% for the first underlayer [2,12,13] due to inaccuracy in the method for complex samples. Additionally, very often the thickness and composition of the underlying layers in multilayer architectures are not exactly known and are introduced in the measurement software using an initial estimate [14].

The challenge of this work is to reduce this source of error in the results and their dependence on standards by introducing a new semi-quantitative method (only the pure element spectra will need to be measured) based on Monte Carlo (MC) simulations. MC simulate X-ray spectra using a statistical approach that counts the photon interactions in the sample. With this approach, inhomogeneities of the sample, spectral and spatial distribution of the beam, polarization effects, photo-absorption, multiple fluorescence, and scattering effects can be considered. Thickness gauging using the MC method is already reported in the literature; most of the reported works are in the field of cultural heritage applications [15–19]. In these cases, simulations are compared with the experimental measurement to confirm hypotheses based on bulk chemical composition, structural observations, and historical information.

The approach described in this article differs from the state of the art in the sense that we use simulations to build calibration curves to determine the thickness of the coating. The same calibration curve could be used for many samples instead of performing many simulations based on hypotheses. This concept will be particularly interesting for industrial applications in metal deposition factories. The simulations require a fast MC code, which is presently part of two software programs for such applications: XRMC [20] and XMI-MSIM [21,22]. Both codes use the Xraylib database [23]. XRMC is generally used for complex 3D geometries since XMI-MSIM can only simulate samples composed of parallel layers. However, we decided to use XMI-MSIM because in our case the geometry is simple, and this program is currently superior to XRMC in simulating XRF experiments [20]. XMI-MSIM is the successor of MSIM, with a history of improvements of over 25 years [24–27].

In this work we examined a single-layer sample of Au, Pd, Sn, and white bronze on brass, using both certified single-element coatings and electroplated alloys. The results were compared with other techniques for data validation: FP, FP + single empirical point, and scanning electron microscopy equipped with a focused ion beam (FIB-SEM). This is expected to provide an analytical method to determine the thickness of coatings that does not make use of standards and whose performance is comparable to or even better than that of XRF analysis with energy-dispersive (ED-XRF) systems on metallic coatings. Finally, we varied some parameters in the simulation to find the ones that are critical for the measurement and those that can be neglected to obtain reliable results.

## 2. Materials and Methods

The metal substrate consists of 3.75 × 5 mm$^2$ brass (copper-zinc alloy) plates, 0.25 mm thick. The substrate was electroplated with palladium and gold using a commercial galvanic bath "720 PDFE MPM" and "8693 MUP" from Bluclad srl (Prato, Italy) and white brass bath "SCUDO BIANCO PLUS RACK" from MacDermid (Waterbury, CT, USA). The alloy composition and layer thickness of the coatings are the subject of this study, and thus they will be discussed later. Certified samples with known thickness are also used and were provided by Bowman (Schaumburg, IL, USA).

XRF measurements were performed with a Bowman B Series XRF spectrometer (Schaumburg, IL, USA) using an acquisition time of 60 s, 50 kV tube voltage, 0.8 mA tube current, and a collimator of 0.305 mm in diameter. The same spectra were used to obtain the thickness information with various methods: FP, FP with one empirical point correction (both available with the commercial software of the instrument), and the MC method proposed in this study.

The composition of the substrate and the coatings were measured with energy-dispersive X-ray spectroscopy (EDS) microanalysis, applying an accelerating voltage of 20 kV and scanning an area of approximately 0.1 mm$^2$ for a live time of 120 s. In order to consider the matrix effect, the ZAF correction algorithm (atomic number, absorption, and fluorescence) was used for quantification. For this purpose, a gold-plated, palladium-plated, and white-bronze-plated sample were prepared, whose thicknesses were high enough to be considered infinite for the EDS analysis. The EDS analysis was performed with a Hitachi S-2300 (Tokyo, Japan) equipped with a Thermo Scientific Noran System 7 detector (Waltham, MA, USA) and analyzed with Pathfinder software (version 2.1) [28].

The SEM images and the FIB ablation were performed with a GAIA 3 equipped with a Triglav electron column and a Gallium FIB Cobra Gallium column manufactured by Tescan (Brno, Czech Republic).

XRF spectra simulations were performed with the open-source software XMI-MSIM v7.0 64-bit by Schoonjan et al. [21,22], which predicts the spectral response of ED-XRF using MC simulations. The software allows for setting many variables of the system under investigation as well as the hardware geometry: this information was used as an input to simulate the spectra.

The quantification method consisted of using the simulated spectra of five different layer thicknesses to build a calibration curve, which was used to extrapolate the unknown thicknesses of the measured samples. Simulations were performed using the exact composition of the coatings and the substrates that were measured with EDS. The spectrum of each pure element of interest (Cu, Zn, Pd, Sn, and Au) was also both measured and simulated to obtain the relative intensity of the peak of interest, called the Peak Ratio (PR) henceforth. The PR concept is similar to the K-ratio used in the EDS [29,30] and consists of the ratio between the peak intensity (X-ray counts) for the element of interest in the sample and the peak intensity at the same energy for the pure element (Equation (1)):

$$PR_i = \frac{I_i^{\text{sample}}}{I_i^{\text{pure}}} \tag{1}$$

XRF spectra were interpolated through multiple Gaussian [31–34] functions in the proximity of the energy lines of the expected elements to obtain the peak area. The considered peaks were Cu K$\alpha$, Zn K$\alpha$, Au L$\alpha$, Pd K$\alpha$, and Sn K$\alpha$; in addition, Cu K$\beta$ and Zn K$\beta$ were also fitted to avoid errors due to peak overlaps. The PR were calculated, and the resulting data were fitted with a second-order curve. This kind of function is commonly implemented in XRF systems for industrial applications since it is in good agreement with experimental data for a limited range of thicknesses and is also easy to manage. The complete quantification procedure is summarized in Figure 1.

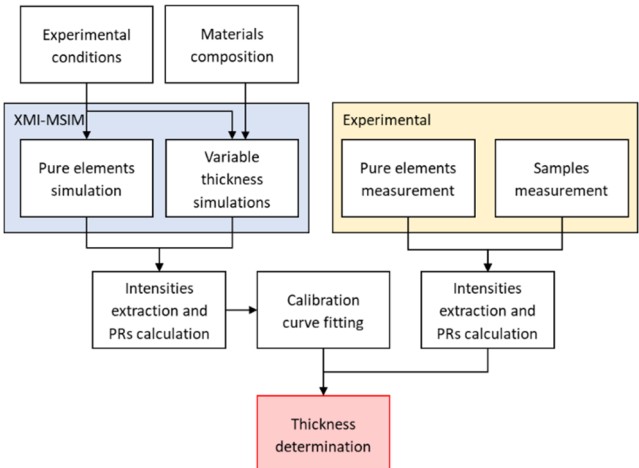

**Figure 1.** Flowchart of the algorithm used for the quantification process.

## 3. Results

### 3.1. Software Validation

The applicability of the proposed method is strongly connected to the ability of the simulation software to provide good results; for this reason, we evaluated the accuracy and the reproducibility of XMI-MSIM.

A parameter that affects the accuracy of the simulations is the number of interactions per trajectory: this number determines the maximum number of interactions that a photon can experience during its trajectory. Low values cause truncation errors, but too high values could result in a computationally expensive simulation without any significant benefits. Simulation of 1 μm of gold coating on brass was performed using values from 1 to 10 as the number of interactions; the PR of each element for all the spectra was compared to the simulation with the highest number of interactions permitted, and the relative deviation was calculated (Figure 2a). The results show an exponential improvement for the first four interactions then, by increasing the number of interactions, the deviation remains stable around 0.001%: for this reason, all the following simulations were performed using four interactions per trajectory.

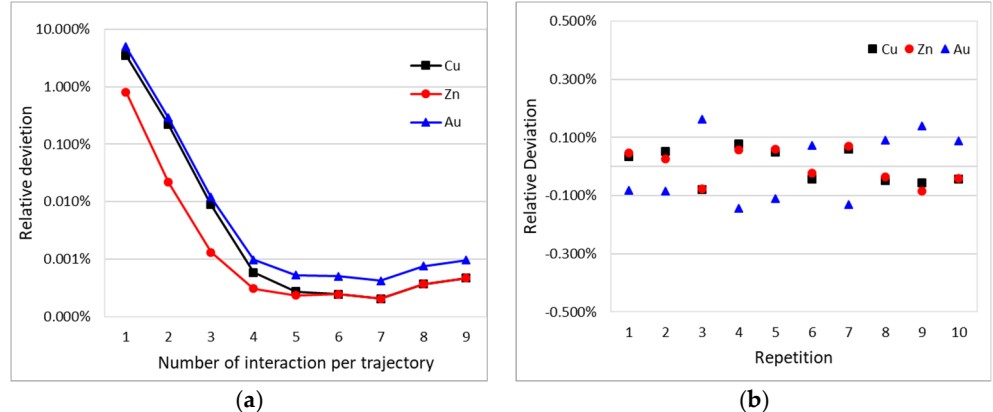

(**a**)                                                             (**b**)

**Figure 2.** PRs of Cu, Zn, and Au of a simulated 1 μm of gold coating on brass substrates: (**a**) relative deviation of the PRs increasing the number of interactions respect to the simulation with 10 interactions; (**b**) relative deviation from the mean value repeating the same simulation 10 times.

The precision of the simulated spectra was evaluated repeating the same simulation on samples consisting of 1 μm of gold on brass substrates 10 times. Then the deviation of the PR of each element from the mean value was calculated (Figure 2b) as well as the relative standard deviation, which is around 0.1%.

After these tests, it can be concluded that the software results are good enough to allow its use in the study and to proceed with the following experiments.

### 3.2. Thickness Determination

After the preparation of the samples, they were measured with the XRF, then the spectra of certified samples and the pure elements Au and Pd were collected. The FP method considers the precious coating as pure for the thickness quantification. It was used both alone and combined with a single empirical point. For the empirical point, the certified calibration standards were used. Then, the thickness of the electroplated sample was measured with FIB-SEM performing a semi-destructive micro-cross section (Figure 3).

A thick deposit of Au and Pd was electroplated separately (approximately 1.1 and 1.9 μm, measured with XRF) and measured with EDS to find the actual composition (Table 1). The composition of the certified thicknesses standards is known and is reported in Table 1 as well. The results agree

with the technical sheets of the baths; except the bronze that show a level of Sn higher (47.2%) than expected (28%–35%), this information will be useful in the determination of the thickness.

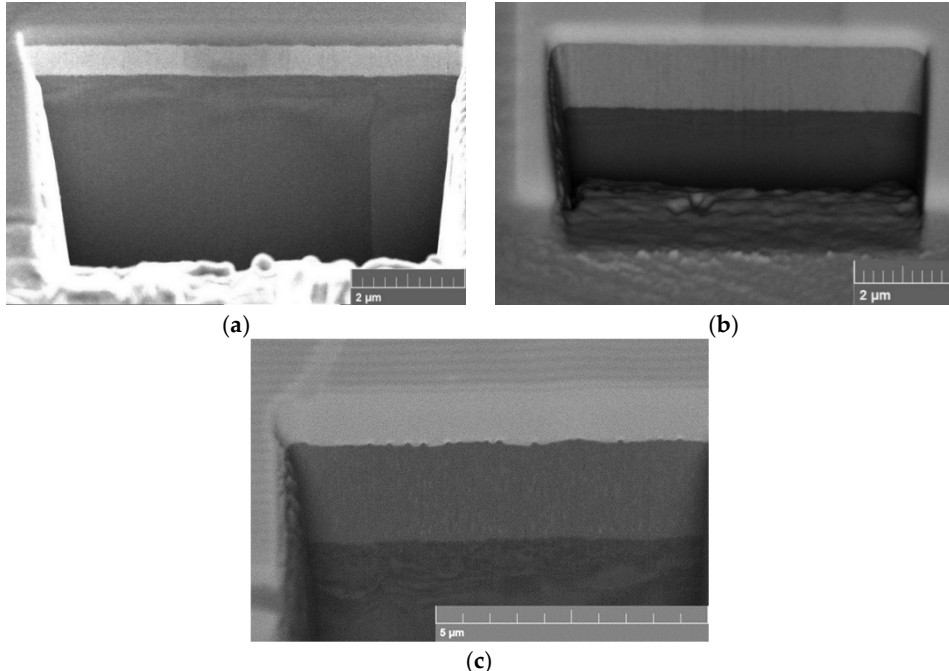

**Figure 3.** FIB-SEM images of the electroplated samples: (**a**) gold; (**b**) palladium; and (**c**) white bronze.

**Table 1.** The composition of the substrate and the film investigated using EDS analysis.

| Samples | Electroplated | Certified |
|---|---|---|
| Brass (substrate) | Cu: 63.0 wt % <br> Zn: 37.0 wt % | Cu: 63.0 wt % <br> Zn: 37.0 wt % |
| Au | Au: 97.9 wt % <br> Fe: 1.6 wt % <br> Ni: 0.5 wt % | Au: 100 wt % |
| Pd | Pd: 95.2 wt % <br> Fe: 4.8 wt % | Pd: 100 wt % |
| Bronze/Sn | Cu: 39.9 wt % <br> Zn: 12.9 wt % <br> Sn: 47.2 wt % | Sn: 100 wt % |

We also performed simulations with XMI-MSIM using the exact concentrations; moreover, the intensities of the peaks were integrated using a multiple Gaussian peak fit. The spectra of the pure elements Au and Pd were also simulated to obtain the PRs. Finally, we obtained six calibration curves (Figure 4), each one containing five points corresponding to different thickness values of the metal: for electroplated gold, certified gold, electroplated palladium, certified palladium, and certified Sn we use 0.1, 0.5, 1.0, 1.5, and 2.0 µm thicknesses, while for electroplated white bronze we use 0.1, 0.75, 1.5, 2.25, and 3 µm, since we expected a thicker coating. The points were fitted with a second-order curve obtaining an $R^2 > 0.9999$.

The peak intensities in the measured spectra were fitted with the same multiple Gaussian curves to calculate PR values and find the thickness of the samples. The results (Table 2) show a large discrepancy in all the samples between the nominal value and the FP method; this deviation is highly improved with the empirical correction. Keeping in mind that the certified samples were also used as standards for the empirical correction, the good results obtained for the certified samples are not very

surprising; moreover, the result in the case of the electroplated samples are improved but still with high accuracy error. On the other hand, the results obtained with the MC method are very promising: the estimated difference is below 2% in four out of six cases, and for two samples the difference (0.0%) is under the precision of the measurements. For the cases of electroplated palladium and bronze the deviation is higher, around 5%, but they are still better than the FP result. The causes that produce these outliers will be studied more deeply in the future, but we can advance hypotheses based on what we observed during the quantification process. The Sn and Pd peaks are not very intense, due to the characteristics of the samples and the detector; in these cases the signal to noise ratio not very high, and for the same reason the matrix effect and the background subtraction are important factors that must be taken into consideration for accurate quantification.

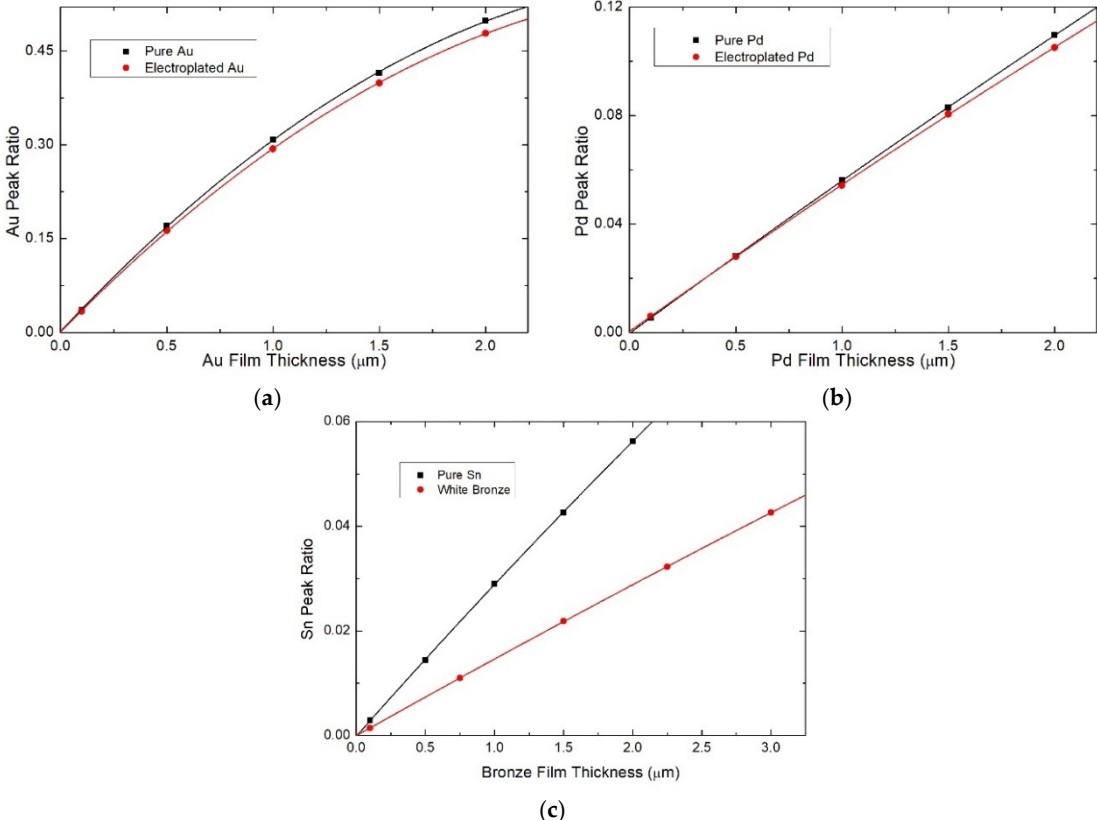

**Figure 4.** Calibration curves built from the simulations using the pure metals (black) and the composition of the electroplated films (red) of (**a**) Au, (**b**) Pd, and (**c**) Sn.

**Table 2.** Nominal (certified and FIB-SEM) and measured (XRF) thickness calculated with FP and MC methods for the samples.

| Samples | Nominal | Experimental | | | | | |
|---|---|---|---|---|---|---|---|
| | | FP (µm) | Difference (%) | FP + Empirical (µm) | Difference (%) | MC (µm) | Difference (%) |
| Certified Au | Certification (µm) | 1.07 | 0.77 | −28.0 | 1.07 | 0.0 | 1.08 | +0.9 |
| Certified Pd | | 1.02 | 0.82 | −19.6 | 1.01 | −1.0 | 1.02 | 0.0 |
| Certified Sn | | 2.08 | 1.42 | −31.7 | 2.01 | −3.4 | 2.04 | −1.9 |
| Electroplated Au | SEM (µm) | 0.53 | 0.28 | −47.2 | 0.39 | −26.4 | 0.53 | 0.0 |
| Electroplated Pd | | 1.30 | 1.15 | −11.5 | 1.40 | +7.7 | 1.23 | −5.4 |
| Electroplated Bronze | | 1.76 | 1.31 | −25.6 | 2.01 | +14.2 | 1.85 | +5.1 |

The fitting of the calibration curve is good enough that if it is repeated by considering only three of the five simulated spectra, the variation will be only approximately ±0.3%, meaning that, in this case, this variation could be acceptable, and the computational cost could be decreased substantially. On the other hand, we found that the film composition influences the results strongly: in Figure 4 there is an appreciable divergence, increasing the thickness, when using pure metal coating standards or the electrodeposited alloy, even if the composition varies only by a few percentage points. If pure standards were used for the quantification of the galvanic sample, the results would have a variation of up to 5%. For instance, the result of FP for the electroplated bronze with the expected Sn concentration in the alloy, was 35%; it would have been 2.71 μm (54% deviation from the real value), and for this reason, we performed the EDS analysis to obtain the exact composition. Unfortunately, it is often assumed that the composition of the deposit does not change much over time, leading to gross errors.

## 3.3. Critical Parameters of Measurement

In the previous section, we proved the power of the MC method for thickness determination. Then, more simulations were performed to predict the critical parameters that we need to take into account when we measure a sample, regardless of the measurement method used.

The first variable we considered was the thickness of the substrate: on too thin samples, the X-rays could pass through, giving a PR different from what is expected. To investigate this phenomenon, we simulated sandwich-like samples with 0.5 μm pure gold coating on both sides (to mimic a real galvanic sample) and a brass layer in between, whose thickness ranged from only 1 μm up to 1 cm, and performed one simulation per order of magnitude. The influence of the thickness of brass substrates is evident only for very thin dimensions (Figure 5a): over 0.1 mm, the difference to an infinite bulk substrate is negligible, and the same calibration curve can be used for different samples.

Later, we investigated the influence of the alloy composition of both the coating and the substrate on the PR. We examined typical deposits of common thickness used in the galvanic industry: a 0.5 μm gold coating alloy, a 3 μm white bronze alloy, and brass alloys. The expected variation in the PR when varying the composition of the alloys depends on the secondary fluorescence and self-absorption of the sample, which depend on the composition, meaning that for different elements the trend could be different.

We investigated the influence of the brass composition on the Au PR. We simulated a 0.5 μm pure gold coating on 1 cm brass substrate. Typical brasses are alloys of Cu and Zn in ratios ranging from 62:38 (UNS alloy number C27200) to 70:30 (C26130) [35]. We simulated the following concentrations of Cu: 63%, 65%, 67%, and 69%. In this range of concentration, the Au PRs does not change significantly remaining in the error of the simulation (Figure 5b); the influence of the substrate could be more remarkable for higher differences in the composition [36].

Then, the effect of the alloy composition of the coating was studied. First, we investigated the Au-Cu alloy from pure gold to 18 kt (75 wt % Au) every 2 kt. As expected, the PR varies with the concentration but, as also found for the electroplated gold with a pure gold calibration curve, the variation is bigger than the variation of gold concentration (Figure 5c); 22 kt gold (8.28 wt % Cu) gives a value of −16.1% with respect to the 24 kt, while 18 kt gold (25 wt % Cu) gives −41.0%. These findings indicate that the results obtained with a not appropriated calibration curve cannot be corrected simply by using a multiplicative factor even if the composition is known. For example, if an 18 kt gold sample is measured and the thickness is estimated with a 24 kt curve to be 0.5 μm, we cannot affirm that the actual thickness is 0.5 μm × 24 kt/18 kt = 0.67 μm since this would underestimate the value.

The same study was carried out with a white bronze alloy on brass. The typical galvanic white bronze composition is 50–55 wt % Cu, 28–32 wt % Sn, and 14–20 wt % Sn. The calibration curve is typically built using a standard of pure Sn, and the results are multiplied by 3.33 (assuming 30 wt % of Sn in the alloy). We simulated an alloy, keeping fixed the amount of Zn at 17% and varying the concentration of Cu and Sn, using Sn concentrations from 28 wt % to 32 wt % every 1 wt %. The simulated sample consists of a 3 μm bronze layer, a 5 μm Cu layer, and 1 cm of brass substrate:

these thicknesses and layer combinations are typical of nickel-free electrodeposition for wearable accessories. Also in this case, similar to what we found for different alloys of gold, the composition plays a crucial role in the thickness determination (Figure 5d) and the approximation, in this case, results in even worse agreement: a variation of just four percentage points leads to an error of 12.2%.

We also simulated a sample with 32 wt % Sn but with 10 µm of the underlying Cu layer, to see if there would be any variation, but the thickness of the layer below did not seem to have a significant influence.

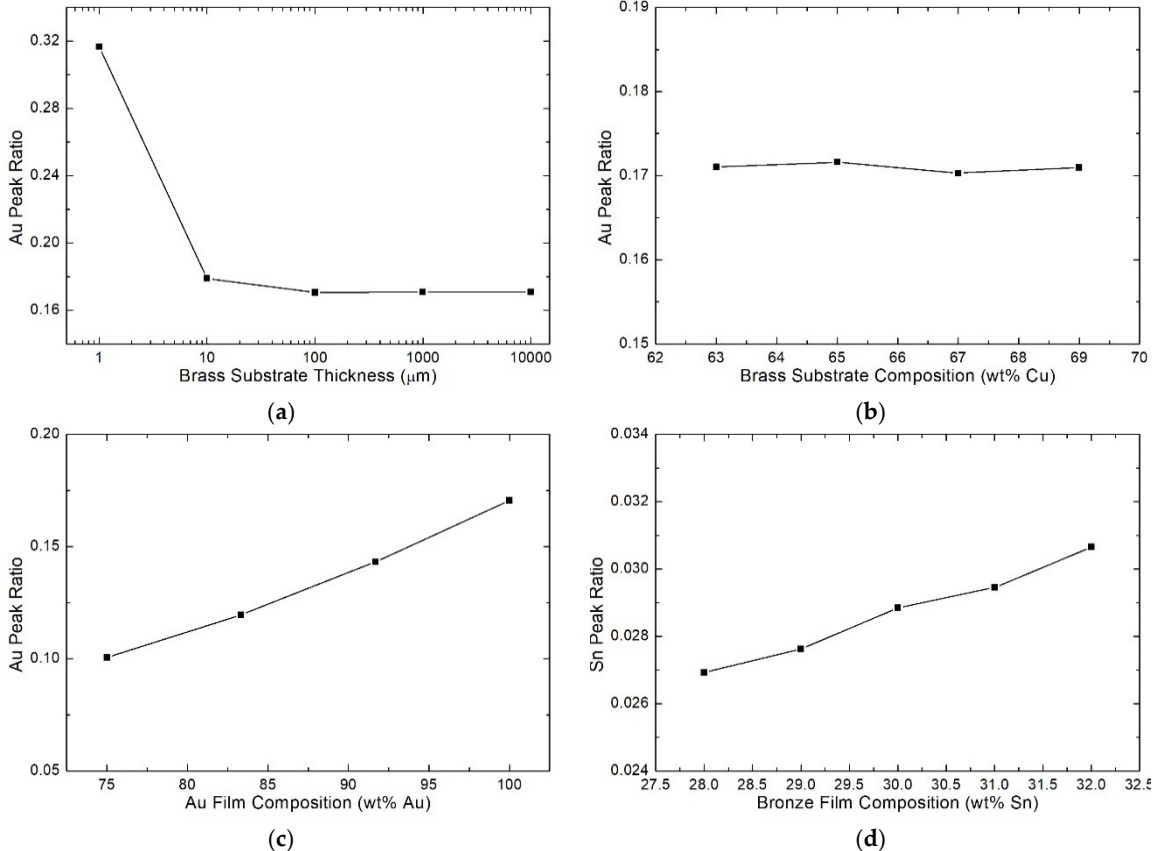

**Figure 5.** Dependence of Au PR of 0.5 µm gold film on brass simulation: (**a**) on the thickness of the substrate; (**b**) on the composition of the substrate (expressed in percentage of Cu); and (**c**) on the composition of the film, increasing the amount of Cu. (**d**) Dependence of Sn PR, of 3 µm bronze film on brass, on the composition of the film, increasing the amount of Cu.

## 4. Conclusions

In the present work, we demonstrated the power of standardless MC calibration for measuring the thickness of metallic coatings. We benchmarked the proposed method with the use of certified standards and SEM imaging of FIB cross sections. Our results indicate that the MC approach competes well with the FP method, which is the state of the art for the measurement of the thickness of metallic coatings in industry. Remarkably, this result was achieved without the use of a standard of known thickness and composition.

The MC method has two major advantages: (i) the lack of standards allows for an easy switch between materials and coating architectures and (ii) it can easily adapt to existing XRF commercial systems as it only requires changes in the software.

After having validated the method and proved that the simulations give reliable results, we have explored a few critical situations that may lead to major errors in the measurement both with the FP and MC methods. We found that in the cases investigated, small variations in the composition and the

thickness of the substrate, or other eventual layers in multilayer architectures, do not play a substantial role in the PR of the element on the top. Conversely, the composition of the substrate alloy must be known exactly; even a small deviation from the known composition can bring about large errors in thickness quantification.

In this work we have used the MC method to investigate the thickness limited to a single layer on a substrate. In the future, we will extend the technique to multilayer structures, introducing a multivariable approach. Such investigation is currently in progress.

**Author Contributions:** Conceptualization, A.L.; formal analysis, W.G.; investigation, W.G. and E.B.; data curation, W.G.; writing—original draft preparation, W.G.; writing—review and editing, W.G. and A.L.; supervision, M.I.; project administration, A.L. and M.I.; funding acquisition, M.I. and A.L.

**Funding:** This research was funded by Regione Toscana POR CreO FESR 2014-2020—azione 1.1.5 sub-azione a1 Bando 2 "Progetti di ricerca e sviluppo delle MPMI," which made possible the project "Gioielli in Argento Da Galvanica Ecologica e Tecnologica" (GADGET) and "Tecnologia al plasma per l'industria del lusso: una manifattura innovativa nel comparto accessori in ottica 4.0" (THIN FASHION). The authors also acknowledge "Ente Cassa di Risparmio di Firenze" Grant Number n. 2013.0878, Regione Toscana POR FESR 2014-2020 for the project FELIX (Fotonica ed Elettronica Integrate per l'Industria), Grant Number 6455.

**Acknowledgments:** Bowman Coating Measurements Systems is gratefully acknowledged for having provided technical information about the hardware and the internal geometry of the instrument. The authors thank the Italian National Research Council (CNR) microscopy facility "Ce.M.E.–Centro Microscopie Elettroniche Laura Bonzi" for providing the facilities for the FIB cross sectioning and the SEM imaging and Carlo Bartoli for the support given in keeping the FIB-SEM up and running.

**Conflicts of Interest:** The authors declare no conflict of interest.

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
