# Peer review of "Coating Thickness Determination Using X-ray Fluorescence Spectroscopy: Monte Carlo Simulations as an Alternative to the Use of Standards"

_coatings, doi:10.3390/coatings9020079_

Reviewer 1 Report

This is a well-presented study that will be very useful in quantifying practical XRF analyses of thin films.

The authors should consider the following suggestions for English language edits which are listed by line number:

53:  on the basis of guess -- using an initial estimate

64:  determine t. -- determine it.

66:  industrial application -- industrial applications

67:  not quite sure what the authors mean, my best guess is:  that is presently owned by two software for such application - that is presently part of two software programs for such applications

76:  focus ion beam -- focused ion beam

76:  It is expected -- This is expected

79:  the ones who are -- the ones that are

80:  measurement and which can be -- measurement and those that can be

86:  the coatings are subject à the coatings are the subject

93:  measured with the energy dispersive -- measured with energy dispersive

95:  ZAF correction algorithm – I don’t think this was defined, please define the acronym

105:  this information have been used as inputs to -- this information was used as input to

116:  in addition to them also Cu -- in addition Cu

116:  were fitted -- were also fit

117:  due to peaks overlapping -- due to peak overlaps

239:  sample consists in a -- sample consists of a

240:  and layers succession -- and layer combinations

242:  brings to even worse results -- results in even worse agreement

256:  was achieved without any standard -- was achieved without the use of a standard

259:  adapt to the existing -- adapt to existing

Author Response

We thank the reviewer for the nice comments and suggestions.

We applied all the proposed variations to make the manuscript more clear and correct in English.

Reviewer 2 Report

Dear authors

This is an excellent work!

I only have two comments that are of minor importance

In line 64 you write : to determine t, is this the thickness?

and on table 1:

The electroplated percentages are the calculated values from XRF analysis? Please clarify on the table.

Author Response

We thank the reviewer for the nice comments on our work.

The issue in line 64 was a typing error, we mean thickness. The compositions were measured using EDS, we have now specified this in the table to clarify.